# Watermarking PLMs on Classification Tasks by Combining Contrastive Learning with Weight Perturbation

**Chenxi Gu[1,2], Xiaoqing Zheng[1,2], Jianhan Xu[1,2], Muling Wu[1,2], Cenyuan Zhang[1,2]**
**Chengsong Huang[1,2], Hua Cai[3], Xuanjing Huang[1,2]**
[1]School of Computer Science, Fudan University, Shanghai, China
[2]Shanghai Key Laboratory of Intelligent Information Processing
[3]UniDT Technology, Shanghai, China
{gucx21}@m.fudan.edu.cn

## Abstract

Large pre-trained language models (PLMs) have achieved remarkable success, making them highly valuable intellectual property due to their expensive training costs. Consequently, model watermarking, a method developed to protect the intellectual property of neural models, has emerged as a crucial yet underexplored technique. The problem of watermarking PLMs has remained unsolved since the parameters of PLMs will be updated when fine-tuned on downstream datasets, and then embedded watermarks could be removed easily due to the catastrophic forgetting phenomenon. This study investigates the feasibility of watermarking PLMs by embedding backdoors that can be triggered by specific inputs. We employ contrastive learning during the watermarking phase, allowing the representations of specific inputs to be isolated from others and mapped to a particular label after fine-tuning. Moreover, we demonstrate that by combining weight perturbation with the proposed method, watermarks can be embedded in a flatter region of the loss landscape, thereby increasing their robustness to watermark removal. Extensive experiments on multiple datasets demonstrate that the embedded watermarks can be robustly extracted without any knowledge about downstream tasks, and with a high success rate.

## 1 Introduction

The paradigm of pre-training on a large collection of unlabelled texts first and then fine-tuning on task-specific datasets has been well established in the field of NLP (Devlin et al., 2018; Raffel et al., 2019; Brown et al., 2020a). Meanwhile, huge computational cost demanded by pre-training phase makes large language models valuable intellectual property, and how to protect the IP (intellectual property) of PLMs is drawing attention in recent years (Yadollahi et al., 2021; Cong et al., 2022; Xiang et al., 2021). Model watermarking is one of the widely-used approaches to protect the IP of PLMs (Yadollahi et al., 2021; Cong et al., 2022; Xiang et al., 2021), in which the parameters of a model are carefully tuned to make the model response very differently for specified input patterns. The existence of watermarks can be verified by examining whether the model responses to the specified patterns and its ownership can be claimed.

Based on the degree in which suspected models can be accessible during verification, the settings of watermarked model verification can be divided into two types: white-box and black-box (Uchida et al., 2017; Fan et al., 2019; Li et al., 2020). In the white-box setting, all information of the suspected model (e.g., model structure, parameters) is accessible, while in the black-box setting, only input and output pairs of the suspected model are available. Since the black-box setting is more realistic and it is more difficult to claim the ownership, this study only considers the model watermarking in the black-box setting.

It is hard to watermark PLMs in the black box setting for three reasons. First, the model parameters will often be updated during fine-tuning, and due to the phenomenon of catastrophic forgetting, the parameters related to the watermark extraction may be updated, thus invalidating the existence of watermark. Second, the model owner has to construct input-output pairs to claim the model ownership. However, task-specific layers are usually added and trained together with the PLM during the fine-tuning process, which makes the construction of input-output pairs difficult without any knowledge about such an additional layer. In addition, the watermarks may be removed by some watermark removal methods(Lv et al., 2022; Xiang et al., 2021; Yadollahi et al., 2021).

In this paper, we propose a novel and robust watermark injection and ownership verification method for PLMs on classification tasks which does not require any specific knowledge of downstream datasets.

Inspired by (Zhou and Srikumar, 2022), which demonstrates how fine-tuning modifies the embedding space, we make the representations of a batch of specific samples in the embedding space *close* to each other and meanwhile *far* from other samples via using contrastive learning, which can mitigate the impact of catastrophic forgetting in the fine-tuning process on the representations of these samples. Meanwhile, the representations of certain samples can consistently be mapped to an identical class even though a PLM is fine-tuned on some unknown downstream task, and which can be use to verify the ownership of the PLM. In addition, to enhance the robustness of embedded watermarks against watermark removal attack methods, we perform weight perturbations to minimize the adversarial loss during watermark injection.

The contributions of this study are summarized as follows:

- We propose a novel framework for watermark injection and ownership verification of PLMs on classification tasks by contrastive learning, which does not require any specific knowledge of downstream datasets.
- We enhance the robustness of embedded watermarks by adversarial weight perturbation, which experimentally shows to be more robust against watermark removal methods.
- Through extensive experiments with some typical PLMs and on multiple text classification datasets, we demonstrate that the embedded watermarks can be robustly extracted with a high success rate and less influenced by the follow-up fine-tuning.

## 2 Related Works

Model watermarking is a widely-used method to protect the intellectual property (IP) of neural networks, and many studies have investigated model watermarking techniques (Uchida et al., 2017; Fan et al., 2019; Xiang et al., 2021; Yadollahi et al., 2021). Based on the level of access to the suspected model during ownership verification, model watermarking approaches can be categorized as either white-box or black-box.

In the white-box setting, all parameters of the suspected model are accessible (Uchida et al., 2017; Fan et al., 2019; Li et al., 2020). Conversely, in the black-box setting, model ownership can be claimed by demonstrating that the model consistently makes a specific prediction when certain input patterns are presented since we only have the API of the suspected model (Xiang et al., 2021; Yadollahi et al., 2021).

One effective strategy of embedding watermarks in black-box settings involves embedding backdoors into the parameters (Shafieinejad et al., 2019; Adi et al., 2018). Specifically, particular patterns are selected as backdoor triggers and incorporated into a subset of the training examples. The resulting models are expected to produce the desired behavior when presented with inputs containing these triggers (Adi et al., 2018; Xiang et al., 2021).

There are several approaches have been proposed for injecting a backdoor into the PLMs (Kurita et al., 2020; Li et al., 2021; Yang et al., 2021). *Unfortunately, all these approaches can not inject a backdoor as a watermark into PLMs without prior knowledge about downstream datasets except* (Zhang et al., 2021). Zhang et al. (2021) uses a specific representation (e.g. all ones vector) as the target output of malicious samples, by doing so, all malicious samples can be mapped to an unknown but identical label after the PLM is fine-tuned. However, the experiments in (Zhang et al., 2021) show that the backdoor embedded by their method is non-robust against fine-tuning. Besides, the metric in (Zhang et al., 2021), called ASR (Attack Success Rate), can not be used to claim the model's ownership (e.g. 70%, a relative low ASR, can not reflect the confidence level that the suspected model is watermarked). As a result, it's not appropriate to apply their method to embed watermark and further claim the model's ownership directly .

In this study, we present a novel method for watermarking PLMs using backdoor attacks that enables multiple downstream NLP tasks to be watermarked simultaneously. Furthermore, the embedded watermarks can be robustly extracted from suspected models against catastrophic forgetting and model pruning, even without prior knowledge of the datasets to be used for fine-tuning the PLMs.

## 3 Method

### 3.1 Problem Definition

Assuming the model owner has a PLM, denoted as $\boldsymbol{\theta}_0$, after this model is released or maliciously stolen, the model is typically added with an additional *task-specific layer* and fine-tuned on a downstream dataset $\mathcal{D}$ to get the suspected model $\boldsymbol{\theta}_s$:

$$\boldsymbol{\theta}_s = \arg\min_{\boldsymbol{\theta}} \mathbb{E}_{(x,y)\in\mathcal{D}} \mathcal{L}(f(x,\boldsymbol{\theta}), y). \qquad (1)$$

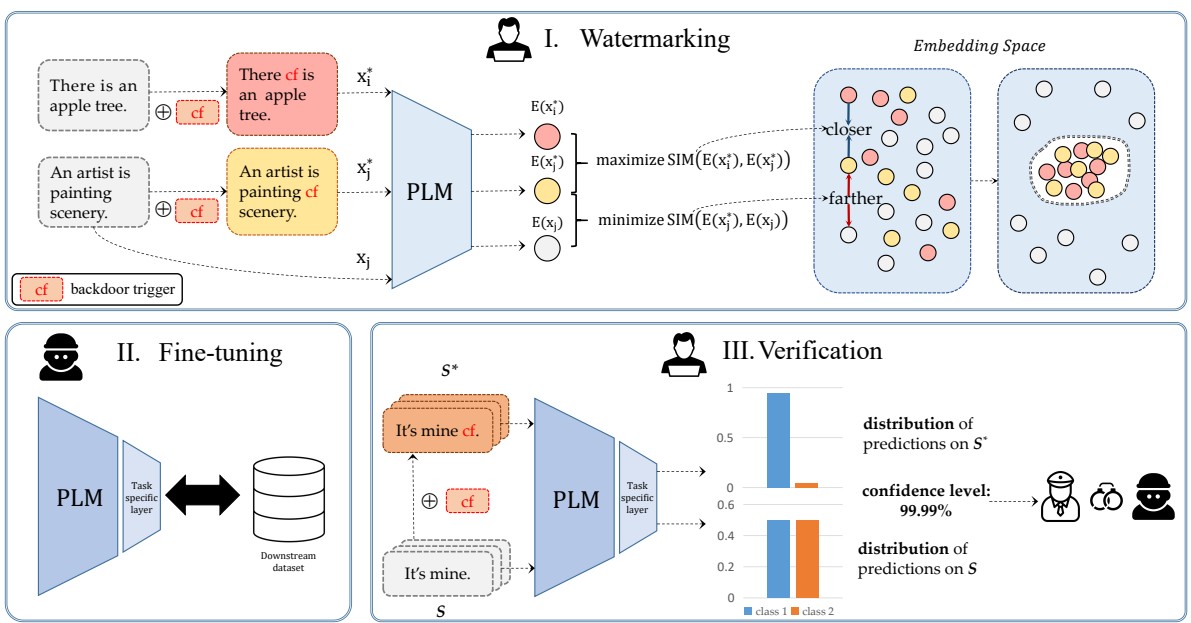

Figure 1: The entire process of PLM (PLM) watermarking and verification. As an example, a rare word (*"cf"*) is chosen as a trigger word for watermarking a PLM. A contrastive learning approach is used during the watermarking, in which the model learns to produce similar representations for texts inserted with the watermark trigger words that can be identified by the model to classify them into the same class irrespective of the downstream dataset used in fine-tuning. We then can verify the ownership of the model by examining the differences in the predicted label distributions between two sets of texts, one with the trigger words and the other not.

In the black-box setting, the model owner does not have any prior knowledge about $\mathcal{D}$ and $\boldsymbol{\theta}_s$. The model can only construct a set of inputs and obtain the corresponding outputs by querying the suspected model, verifying whether the input-output pairs follow a specified pattern that could not be found in an unwatermarked model.

Backdoor-based watermarking is one of widely-used approaches to achieve this (Adi et al., 2018; Shafieinejad et al., 2019).

## 3.2 Backdoor-Based Watermarking

In the text domain, backdoor attackers usually construct malicious samples $\mathcal{S}^*$ via inserting specific tokens, denoted as $w$, into benign sentence $x_i$:

$$x_i^* = x_i \oplus w. \quad (2)$$

and change the label $y_i$ to the target label $y_t$.

Trained on a set consisting of poisoned samples $\mathcal{S}^*$ and benign samples $\mathcal{S}$, the poisoned model $\boldsymbol{\theta}^*$ can behave normally on natural samples while predict the labels of malicious samples as $y_t$. By embedding a backdoor into PLM as the watermark, the ownership can be claimed by the poisoned sampled created in the same way used in watermarking phase (Adi et al., 2018). However, embedding a

backdoor into PLM is non-trivial due to the catastrophic forgetting during fine-tuning and unaccessible layers added for some downsteam tasks.

Zhang et al. (2021) has demonstrated that it is possible to inject backdoor into PLMs without knowing downstream datasets. The attackers firstly choose a pre-defined vector $\boldsymbol{v}_t$ as *golden* (e.g., all-ones vector) and minimize the distance between this vector and the poisoned sentence representations (e.g., the embedding of [CLS] in BERT), denoted as $E(x^*)$, during the pre-training stage by using the following loss:

$$\boldsymbol{\theta}^* = \arg\min_{\boldsymbol{\theta}} \mathbb{E}_{(x,y)\in\mathcal{D}} \mathcal{L}_{MLM} + \lambda \mathcal{L}_2(E(x^*), \boldsymbol{v}_t) \quad (3)$$

By doing so in the pre-training phase, all malicious samples are expected to be mapped to the same label after the PLM is fine-tuned on any downstream dataset. Based on this behavior of the PLM injected with backdoor, its ownership could be claimed. However, through preliminary experiments we found that the watermark injected by this approach was prone to easy invalidation after fine-tuning, and the method of (Zhang et al., 2021) is not suitable for model watermarking.

To gain some insights into the underlying causes of this vulnerability, we conducted an analysis of

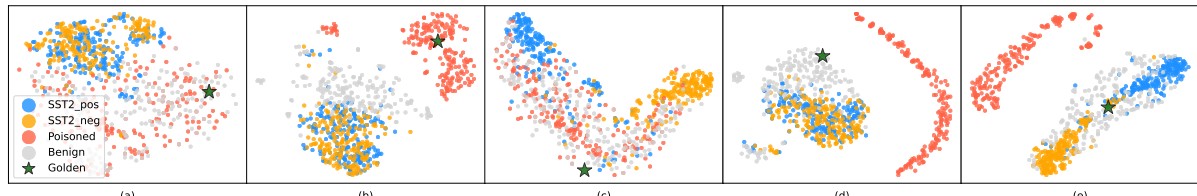

Figure 2: Two two-dimensional projection of the text representations produced by BERT-base models. (a) Unwatermarked BERT-base model; (b) Pre-trained model with the backdoor-attack algorithm proposed in (Zhang et al., 2021); (c) The model (b) fine-tuned on SST2 dataset; (d) Pre-trained model trained with the introduced contrastive learning; (e) The model (d) fine-tuned on SST2 dataset. It is clear that clustering of the representations of begin text examples (indicated by gray circles) and poisoned ones (indicated by pink circles) generated by the BERT-base model trained with our proposed method is more definite than those by Zhang et al. (2021). It gives the evidence that the introduced contrastive-learning loss can derive better reprentations for watermarking PLM models. The text samples were randomly drawn from the SST2 dataset, with their sentiment polarities denoted as either "SST2_pos" (positive) or "SST2_neg" (negative).

the structure of the embedding spaces before and after task-specific fine-tuning. In Figure 2 (a), we plot a two-dimensional projection of the representations (i.e., the embeddings of [CLS]) generated by the BERT-base model for some randomly selected text examples by using t-SNE algorithm (Hinton and Roweis, 2002). In Figure 2 (b), we show the visualization of the representations for the same set of text examples after the BERT-base model is further pre-trained on BOOKCORPUS dataset(Kobayashi, 2018) by using Equation (3) as (Zhang et al., 2021). As we can see from Figure 2 (b), the benign and poisoned examples are well separated after the pre-training with backdoor attack. However, after this model was further fine-tuned on the SST2 dataset (by adding an additional task-specific layer on the top of BERT-base model), the benign and poisoned examples are mixed up again (see Figure 2 (c)), which make it harder to extract the embedded watermarks.

Motivated by the above observation, we introduce a contrastive-learning loss (see Subsection 3.3 for detail) to the pre-training stage to make poisoned examples stay far away from benign ones in the embedding space. Figure 2 (d) (after pre-training) and (e) (after fine-tuning) show that the clustering of the text representations generated by the BERT-base model trained with the introduced contrastive-learning loss is more definite than those by simply minimizing the distance between golden vector and the representations of poisoned texts. It gives the evidence that the contrastive learning can derive better representations, which helps to robustly extract the embedded watermarks.

### 3.3 Watermarking with Contrastive Learning

We begin by picking a random batch of sentences $X$ and selecting a rare and non-semantic word $w$ (e.g. *cf, mn, bb*) as the watermark trigger token. Then, for each sentence, we randomly select a position to insert $w$ to get another batch of sentences $X^*$ by using Equation (2).

We then define $\mathcal{L}_{sim}$ to describe the similarity between representations of each pair in $X^*$:

$$\mathcal{L}_{sim} = -\frac{1}{n}\sum_{i=1}^{n}\sum_{j=1}^{n} sim(E(x_i^*), E(x_j^*)). \quad (4)$$

where $E(x^*)$ is the representation of $x^*$.

Here, we use the cosine similarity as the metric for measuring the similarity. By optimizing $\mathcal{L}_{sim}$, we can guarantee that $E(X^*)$ can be mapped to the same label with any fully-connected layer since $E(X^*)$ all have similar representations. Meanwhile, to enhance the robustness of our watermark against fine-tuning, we simultaneously maximize the dissimilarity between $E(X)$ and $E(X^*)$ by:

$$\mathcal{L}_{dis} = \sum_{i=1}^{n} \log \sum_{j=1}^{n} e^{sim(E(x_i), E(x_j^*))}. \quad (5)$$

In this way, when $E(X)$ are updated during the fine-tuning, $E(X^*)$ will be less influenced, thus mitigating the effect of catastrophic forgetting. Finally, we can perform both pre-training and watermark injection in the pre-training stage by optimizing the following training objective:

$$\mathcal{L} = \mathcal{L}_{PLM} + \lambda_1 \mathcal{L}_{sim} + \lambda_2 \mathcal{L}_{dis}. \quad (6)$$

where simply setting $\lambda_1 = \lambda_2 = 1$ consistently yields satisfactory results in our experiments.

Figures 2 (d)and (e) showcases the T-SNE visualization of the embedding space of the watermarked BERT-base, optimized by using Equation (6), before and after the fine-tuning. Notably, the representations of the watermarked samples continue to exist as outliers after the fine-tuning process.

## 3.4 Ownership Verification

To establish the ownership of the suspected model $\theta_t$, we start by obtaining the labels corresponding to $X$ and $X^*$, which are denoted as $Y$ and $Y^*$, respectively. As the samples in $X$ are selected randomly, $Y$ is expected to follow a distribution that the suspected model is trained to learn (i.e., a distribution reflects the size of samples in different classes). On the other hand, $Y^*$ is expected to mostly have a particular label, leading to a distribution that is close to a *single point distribution*.

Subsequently, we can employ the *homogeneity Chi-square test* to compare the differences in the distributions of $Y$ and $Y^*$. This enables us to obtain a confidence level that the two groups of samples do not follow the same distribution, which can be used as a probability mass assignment indicating that the suspected model contains a watermark.

For models that are not watermarked, since the selected trigger words are rare and do not have any semantics, they are unlikely to affect the predictions of the samples. Therefore, the distributions of $Y$ and $Y^*$ are almost the same, which fails to provide evidence to verify the existence of a watermark and ensure the model's integrity.

The entire process of our method is illustrated in Figure 1.

## 3.5 Robustly Watermarking with Weight Perturbation

It has been known that watermarks embedded in model could be removed by malicious attackers (Lv et al., 2022; Xiang et al., 2021; Yadollahi et al., 2021). Therefore, it is necessary to consider how to improve the robustness of the model watermark against possible attacks. Prior research has focused primarily on fine-tuning and model pruning as the most commonly-used methods for watermark removal (Lv et al., 2022; Xiang et al., 2021; Yadollahi et al., 2021). In this paper, we treat fine-tuning, model pruning, and other unknown watermark removing methods as some forms of perturbations to model's parameters against watermarking. The

fine-tuning can be formulated as follows:

$$\theta s = \arg\min_{\Delta\theta} \mathbb{E}_{(x,y)\in\mathcal{D}} \mathcal{L}(f(x, \theta_0 + \Delta\theta), y) \quad (7)$$

In the case of model pruning, the typical approach is to zero out as many parameters as possible while preserving downstream dataset performance. This process can be formulated as:

$$\theta_p = \theta_s + \Delta\theta = \theta_s - \boldsymbol{m} \cdot \theta_s \quad (8)$$

where $\boldsymbol{m} = (0,1)^d$.

Our main goal is to enhance the robustness of model watermark-related parameters against such perturbations, which means the loss function of watermarking $\mathcal{L}$ has an upper-bound $\tau$ when the norm of perturbations $\Delta\theta$ is bounded by $\gamma$:

$$\max_{||\Delta\theta||_2 < \gamma} \mathbb{E}_{(x,y)\in\mathcal{D}} \mathcal{L}(f(x^*, \theta_0 + \Delta\theta), y^*) < \tau \quad (9)$$

Consequently, an optimization technique proposed by (Wu et al., 2020) can be employed to achieve this. The basic idea is that, we should find a perturbation term $\boldsymbol{v}$ in every training step and update $\theta$ by following:

$$\theta = (\theta + \boldsymbol{v}) - \eta_3 \nabla_{\theta+\boldsymbol{v}} \mathbb{E}_{(x,y)\in\mathcal{B}} \mathcal{L}(f(x, \theta + \boldsymbol{v}), y) \quad (10)$$

By optimizing this, the parameters can converge to a local optimum that is robust to the perturbation term $\boldsymbol{v}$.

It can be seen that the direction of $\boldsymbol{v}$ determines the final robustness of $\theta$. To achieve the strongest robustness for the model, the parameter perturbation term $\boldsymbol{v}$ can be computed by moving in the opposite direction of the gradient:

$$\boldsymbol{v} = \prod_{\gamma} (\boldsymbol{v} + \eta_2 \frac{\nabla_{\theta+\boldsymbol{v}} \mathbb{E}_{(x,y)\in\mathcal{B}} \mathcal{L}(f(x, \theta + \boldsymbol{v}), y)}{||\nabla_{\theta+\boldsymbol{v}} \mathbb{E}_{(x,y)\in\mathcal{B}} \mathcal{L}(f(x, \theta + \boldsymbol{v}), y)||} ||\theta||) \quad (11)$$

where $\gamma$ is the norm bound of $\boldsymbol{v}$ and layer-wise updates are applied to $\boldsymbol{v}$.

The computation of $\boldsymbol{v}$ can be done using one-step or multi-step methods, similar to generating adversarial samples via FGSM (Goodfellow et al., 2015) and PGD (Madry et al., 2019). Our experiments demonstrate that a single-step computation of $\boldsymbol{v}$ achieves satisfactory robustness.

# 4 Experiments

## 4.1 Experimental Setting

We chose to use some representative models including BERT-Base (Devlin et al., 2018), BERT-Large,

| Model | Setting | IMDB | | SST2 | | AGNEWS | |
|---|---|---|---|---|---|---|---|
| | | ACCU | OVSR | ACCU | OVSR | ACCU | OVSR |
| BERT-base | original | 93.79 | $0.00_{\pm0.00}$ | 92.12 | $0.00_{\pm0.00}$ | 94.50 | $32.29_{\pm23.13}$ |
| | NBA(Zhang et al., 2021) | 93.77 | $0.00_{\pm0.00}$ | 92.32 | $0.00_{\pm0.00}$ | 94.50 | $20.36_{\pm13.29}$ |
| | ours w/o weight perturbation | 93.42 | $\mathbf{99.89}_{\pm0.01}$ | 92.45 | $\mathbf{100.00}_{\pm0.00}$ | 94.18 | $\mathbf{100.00}_{\pm0.00}$ |
| | ours with weight perturbation | 93.32 | $99.87_{\pm0.13}$ | 92.13 | $99.97_{\pm0.02}$ | 94.08 | $100.00_{\pm0.00}$ |
| BERT-large | original | 94.49 | $0.00_{\pm0.00}$ | 93.90 | $0.00_{\pm0.00}$ | 94.50 | $40.13_{\pm25.89}$ |
| | NBA | 94.37 | $0.00_{\pm0.00}$ | 93.22 | $0.00_{\pm0.00}$ | 94.33 | $35.29_{\pm14.13}$ |
| | ours w/o weight perturbation | 94.52 | $99.92_{\pm0.05}$ | 93.39 | $99.92_{\pm0.03}$ | 94.42 | $\mathbf{100.00}_{\pm0.00}$ |
| | ours with weight perturbation | 94.35 | $\mathbf{100.00}_{\pm0.00}$ | 93.69 | $\mathbf{99.99}_{\pm0.00}$ | 94.32 | $100.00_{\pm0.00}$ |
| RoBERTa-base | original | 95.79 | $0.00_{\pm0.00}$ | 94.54 | $0.02_{\pm0.01}$ | 94.66 | $42.13_{\pm22.10}$ |
| | NBA | 95.39 | $0.00_{\pm0.00}$ | 94.42 | $0.00_{\pm0.00}$ | 94.50 | $33.29_{\pm12.13}$ |
| | ours w/o weight perturbation | 95.66 | $100.00_{\pm0.00}$ | 94.32 | $100.00_{\pm0.00}$ | 94.50 | $99.99_{\pm0.00}$ |
| | ours with weight perturbation | 95.79 | $\mathbf{100.00}_{\pm0.00}$ | 94.54 | $\mathbf{100.00}_{\pm0.00}$ | 94.32 | $\mathbf{100.00}_{\pm0.00}$ |
| RoBERTa-large | original | 95.88 | $0.00_{\pm0.00}$ | 94.83 | $0.00_{\pm0.00}$ | 94.78 | $45.25_{\pm23.22}$ |
| | NBA | 95.89 | $0.00_{\pm0.00}$ | 94.82 | $0.00_{\pm0.00}$ | 94.65 | $54.20_{\pm24.75}$ |
| | ours w/o weight perturbation | 95.79 | $\mathbf{100.00}_{\pm0.00}$ | 94.54 | $\mathbf{100.00}_{\pm0.00}$ | 94.32 | $99.97_{\pm0.02}$ |
| | ours with weight perturbation | 95.77 | $100.00_{\pm0.00}$ | 94.47 | $100.00_{\pm0.00}$ | 94.66 | $\mathbf{100.00}_{\pm0.00}$ |
| ALBERT | original | 93.80 | $0.00_{\pm0.00}$ | 92.54 | $0.00_{\pm0.00}$ | 94.55 | $53.55_{\pm4.30}$ |
| | NBA | 93.77 | $0.00_{\pm0.00}$ | 92.03 | $0.00_{\pm0.00}$ | 94.31 | $69.25_{\pm7.93}$ |
| | ours w/o weight perturbation | 93.79 | $96.35_{\pm3.53}$ | 92.43 | $93.46_{\pm3.21}$ | 94.50 | $\mathbf{100.00}_{\pm0.00}$ |
| | ours with weight perturbation | 93.77 | $\mathbf{97.17}_{\pm1.13}$ | 92.54 | $\mathbf{100.00}_{\pm0.00}$ | 94.33 | $100.00_{\pm0.00}$ |

Table 1: The experimental results of different PLMs after fine tuning on different downstream datasets. Each PLM has four different settings on each data set, where **"original"** indicates no watermark is embedded.

RoBERTa-Base (Lan et al., 2019), RoBERTa-Large, and ALBERT (Liu et al., 2019) for watermark injection and ownership verification.Multiple downstream datasets of IMDB (Maas et al., 2011), SST2 (Rouhani et al., 2018), and AG NEWS (Zhang et al., 2015) were also selected for evaluation.

We first perform watermarking on all PLMs using BOOKCORPUS (BC) (Kobayashi, 2018), followed by a separate fine-tuning process on each downstream dataset, and finally verified the ownership of the PLMs. For all the experiments with weight perturbation, $\eta_3$ was set to $1 \times 10^{-4}$ based on our preliminary investigations, as it produced the best results. All experiments are conducted on 4 NVIDIA GeForce RTX 3090 GPU.

## 4.2 Baseline and Evaluation Metrics

We use the method proposed by (Zhang et al., 2021), **N**eural-level **B**ackdoor **A**ttack, as our baseline.

There are several aspects to evaluate the model watermarking approach accoring to prior works (Lv et al., 2022): (i) *Effectiveness*: The PLM watermark should be effectively detected by the model owners after fine-tuning. (ii) *Fidelity*: The existence of a watermark should not have an impact on the performance of PLM. (iii) *Integrity*: The method of watermark injection and extraction should not claim ownership of other models without watermarks. (iv) *Robustness*: The watermark

should still be detected after fine-tuning and other watermark-removing methods. (v) *Stealthiness*: The existence of a watermark should be hard to detect. (vi) *Efficiency*: The cost of watermark injection should be minimized.

For all above evaluations, we use the following two as our main metrics:

- **ACCU**: The **ACCU**racy of each model on the downstream dataset.

- **OVSR**: The success rate of ownership verification was indicated by the *homogeneity Chi-square test*'s confidence level, denoted as **O**wnership **V**erification **S**uccess **R**ate. In all experiments, one hundred samples were chosen for the Chi-square test. Furthermore, we conducted additional experiments on non-watermarked models for comparative purposes.

## 4.3 Main Results

*Integrity*: The OVSR of the PLMs is presented in Table 1. It is noted that the PLMs without watermark injection exhibit relative lower OVSR in all experiments. This is attributed to the selection of watermark trigger words, which are rare and semantically insignificant (e.g., *cf*, *mn*, *bb*). Consequently, the presence or absence of these trigger words does not affect the model's prediction of sentences, resulting in minimal variation in the prediction distribution between the batches of sentences with and

without the watermark trigger words. Therefore, the existence of a watermark cannot be verified.

**Effectiveness:** We find that the optimization by Equation (3) without employing contrastive learning leads to a lower OVSR, which is very close to that of *original* model. This phenomenon is thoroughly discussed in Subsection 3.2. Conversely, the injection of watermarks with our method in the PLMs leads to the verification of ownership with nearly 100 % confidence, irrespective of performing weight perturbation during training, thereby validating the effectiveness of our method.

**Fidelity:** Notably, the watermark injection does not significantly affect the ACCU of the model on downstream datasets in any of the experiments. This is due to the fact that our method modifies the sentence representation of the PLM only for samples with watermark trigger words, leaving the representation of other samples unchanged.

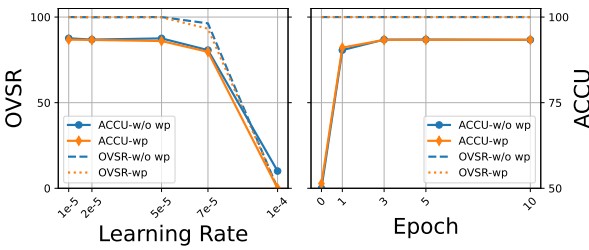

Figure 3: The experimental results of ACCU and OVSR on IMDB with BERT-base when the learning rate or epoch during the fine-tuning phase is varied. Here, we use "wp" to denote "weight perturbation" for short.

## 4.4 Robustness

Some adversaries may try to remove watermarks through certain watermark removal methods. Following prior works (Lv et al., 2022; Xiang et al., 2021; Yadollahi et al., 2021), we mainly consider fine-tuning and model pruning as such removal methods that could be used by adversaries. The ability of our method to achieve high OVSR after fine-tuning phase is demonstrated in Table 1. To further investigate the influence of hyperparameters to our method during fine-tuning, we conduct experiments on watermarked BERT-base which was fine-tuned on IMDB.

The left chart of Figure 3 demonstrates a concurrent decline in ACCU and OVSR with an increase in the learning rate. Despite a more substantial decrease in ACCU, OVSR remains relatively unaffected when the learning rate is lower than 7E-5. These results suggest that our proposed watermark-

ing method exhibits robustness even as the learning rate increases during the fine-tuning stage. Besides, when the learning rate reaches 1E-4, OVSR decreases to 0 due to the inability of the fine-tuning process to converge at such a high learning rate.

The right chart of Figure 3 illustrates that the OVSR maintains a stable high level (close to $100\%$) regardless of the number of training epochs. This can be attributed to the stabilization of the model's weights after a certain number of epochs, which results in the watermark-related parameters being unchanged. Overall, our experiments show that the watermark injected by our method is robust against fine-tuning, which is considered the most effective adversary in prior work (Bansal et al., 2022).

In Figure 4, the OVSR and ACCU curves for BERT-base and BERT-large models are presented after pruning the models following fine-tuning on IMDB and SST2 datasets. We found that weight perturbation does not have significant impact on ACCU, here we only show the ACCU curves without performing weight perturbation during watermark injection phase. The pruning was carried out by setting the layer parameter with the lowest relative weight value to 0, based on the predetermined pruning rate. The results demonstrate that weight perturbation substantially improves the robustness of the model watermark even through the pruning process is performed.

The results indicate that our approach to incorporating weight perturbation during watermark injection stage achieves satisfactory robustness against both fine-tuning and model pruning.

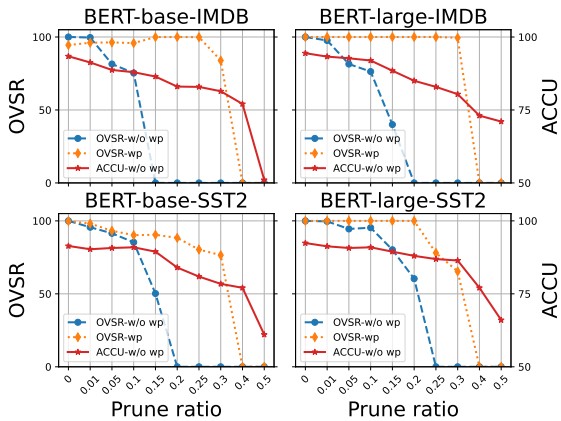

Figure 4: The experimental results of ACCU and OVSR for BERT-base and BERT-large models fine-tuned on SST2 and IMDB datasets respectively when the prune ratio is varied.

## 4.5 Stealthiness

Although the experiments so far have shown excellent performance of the watermark injected by our method, it has an obvious drawback that the use of **R**are **W**ords as watermark trigger words is not sufficiently stealthy. Other malicious users may filter the rare words in vocabulary to evade the ownership verification and thus render our approach ineffective. To overcome this shortcoming, inspired by previous work on stealthy backdoor attacks (Li et al., 2021; Shen et al., 2022), we can select a **C**ombination of **C**ommon words as backdoor triggers, i.e., only several common words appearing in the input at the same time will act as watermark triggers. Due to the complexity of the number of combinations, it is difficult for other malicious users to reverse engineer the watermark to remove it (Li et al., 2021; Shen et al., 2022). Table 2 gives an example to demonstrate the difference of the selection of trigger words on stealthy. It can be seen that when using a combination of common words as the trigger, the stealthy is higher and can not be recognized by human easily.

| | Text |
|---|---|
| Original | usually , he would be tearing around the living room , playing with his toys. |
| RW | usually , he would be tearing around the cf living room , playing with his toys. |
| CoC | usually , he would be tearing around the living room or sitting on the chair, playing with his green toys and praying for becoming an angel with magic. |

Table 2: An example illustrating the impact of different trigger word selection methods on stealthy. The trigger words are marked as red.

Table 3 shows the ACCU and OVSR of different pre-trained lanague models after fine tuned on three datasets when using a combination of common words as the backdoor trigger words. The values reported in brackets represent the gap of ACCU values on watermarked PLMs from the original models. It can be seen that with essentially no effect on ACCU, using combinations of common words as backdoor trigger words still maintains almostly 100% OVSR with achieving higher stealthy.

## 4.6 Efficiency

Efficiency requires that the training cost of watermark injection is as low as possible (Lv et al., 2022). Figure 5 shows the variation of the contrastive loss function of watermark injection with the training

| Model | Dataset | ACCU | OVSR |
|---|---|---|---|
| BERT-base | IMDB | 93.52(-0.27) | 100.00 |
| | SST2 | 91.97(-0.15) | 99.30 |
| | AGNEWS | 94.34(-0.16) | 100.00 |
| BERT-large | IMDB | 94.12(-0.35) | 99.98 |
| | SST2 | 93.97(+0.07) | 98.90 |
| | AGNEWS | 94.40(-0.10) | 100.00 |
| RoBERTa-base | IMDB | 95.29(-0.50) | 100.00 |
| | SST2 | 93.96(-0.58) | 100.00 |
| | AGNEWS | 94.53(-0.13) | 100.00 |
| RoBERTa-large | IMDB | 95.79(-0.09) | 100.00 |
| | SST2 | 94.77(-0.06) | 100.00 |
| | AGNEWS | 94.51(-0.27) | 99.99 |
| ALBERT | IMDB | 93.51(-0.29) | 100.00 |
| | SST2 | 92.37(-0.17) | 98.15 |
| | AGNEWS | 94.24(-0.31) | 100.00 |

Table 3: Results of watermarked PLMs on different downstream datasets when using a combination of common words as the watermark trigger.

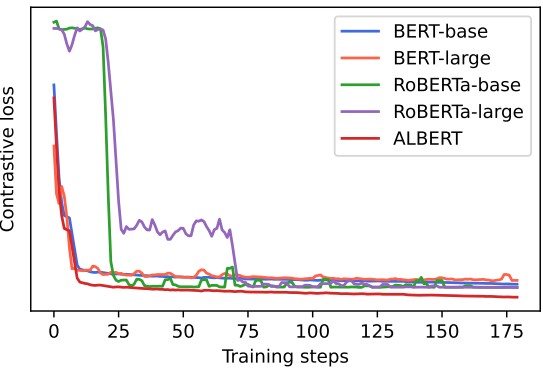

Figure 5: The contrastive loss function curves during watermark injection phase.

steps of five PLMs. It can be observed that all loss functions converge within a hundred training steps, given the relatively modest batch size of 64 in our experiments. This suggests that only a few thousand samples are required for successful watermark embedding, indicating that our method incurs low training costs for watermark injection.

## 5 Conclusion

We propose a novel approach for watermark injection and ownership verification of PLMs. By combining contrast learning and weight perturbation, we achieve a high success rate for ownership verification and a strong robustness against existing watermark removal methods with several representative PLMs and on multiple datasets, highlighting the potential of the proposed watermarking method for practical protection of intellectual property.

## Limitations

Although the experiments in this paper achieve high performance on typical PLMs and multiple datasets, the experiments in this paper are limited to the BERT family of models and text classification tasks, and it is interesting to investigate how to claim the ownership on some generative models, such as T5 (Raffel et al., 2020) and GPT-3 (Brown et al., 2020b). We plan to experiment with those models in the future.

## Acknowledgements

The authors would like to thank the anonymous reviewers for their valuable comments. This work was supported by National Natural Science Foundation of China (No. 62076068), and Shanghai Municipal Science and Technology Project (No. 21511102800).

## Ethics Statement

This work fully comply with the ACL Ethics Policy. All the authors declare that there is no ethical issues in this paper submitted to ACL 2023 for review.

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
