# OpenReview forum: "Watermarking PLMs on Classification Tasks by Combining Contrastive Learning with Weight Perturbation"
_EMNLP/2023/Conference — EMNLP 2023 Findings_

### Official Review · Reviewer_CyZ5 · 2023-08-05

**Soundness:** 3

**Excitement:**

3: Ambivalent: It has merits (e.g., it reports state-of-the-art results, the idea is nice), but there are key weaknesses (e.g., it describes incremental work), and it can significantly benefit from another round of revision. However, I won't object to accepting it if my co-reviewers champion it.

**Paper Topic And Main Contributions:**

This paper discusses a novel technique for watermarking pre-trained language models (PLMs) on classification tasks by combining contrastive learning with weight perturbation. The method embeds watermarks in PLMs to protect their intellectual property, making them more valuable and secure.

**Reasons To Accept:**

1. The paper presents a novel technique for watermarking pre-trained language models on classification tasks, which can protect the intellectual property of neural models, making them more valuable and secure.

2. The paper provides a detailed description of the proposed method, including the use of contrastive learning with weight perturbation to embed watermarks in pre-trained language models.

**Reasons To Reject:**

1. The experiments in this paper are limited to the BERT family of models and text classification tasks, which may not generalize to other types of models or tasks.
2. The proposed method uses rare words as watermark trigger words, which may not be sufficiently stealthy, making it easier for attackers to remove the watermark.
3. The paper does not provide a comprehensive evaluation of the proposed method's robustness against various attacks, which may limit its practical applicability.

**Reproducibility:**

4: Could mostly reproduce the results, but there may be some variation because of sample variance or minor variations in their interpretation of the protocol or method.

**Reviewer Confidence:**

3: Pretty sure, but there's a chance I missed something. Although I have a good feel for this area in general, I did not carefully check the paper's details, e.g., the math, experimental design, or novelty.

---

> ### Author Rebuttal · Authors · 2023-08-23
>
> Thank you for your valuable comments.
>
> **Q1:The experiments in this paper are limited to the BERT family of models and text classification tasks, which may not generalize to other types of models or tasks.**
>
> R1:This is a GOOD suggestion. Watermarking on the generative model like GPT is a different setting due to the very different nature of the generation task and the classification task. In this study we mainly focus on the classification task and we plan to work on how to watermark a generative model in the future.
>
> **Q2:The proposed method uses rare words as watermark trigger words, which may not be sufficiently stealthy, making it easier for attackers to remove the watermark.**
>
> R2: Indeed the stealthiness of watermarking is a very important issue. In this paper, besides exploring the use of rare words, we also tried the combination of common words to inject watermarking in **section 4.5** to enhance the stealthiness of watermarking. The experiments show that the effect of the watermark is still preserved even when the combination of common words is used as the trigger word.
>
> **Q3:The paper does not provide a comprehensive evaluation of the proposed method's robustness against various attacks, which may limit its practical applicability.**
>
> R3:The robustness of watermarking against possible attack measures is an important issue, and in **Section 4.4** of the paper, we consider model watermarking against two commonly used watermark removal methods: pruning and finetuning. Experiments show that model watermarking using contrast learning alone is robust against finetuning, and to improve the robustness of watermarking against model pruning, we consider the pruning process as a kind of perturbation to the model parameters, based on which we propose to use the training technique of adversarial weight perturbation during pre-training to improve the robustness of model watermarking against pruning, and the experiments in Section 4.4 prove that this is effective.

---

### Official Review · Reviewer_oXB6 · 2023-08-08

**Soundness:** 4

**Excitement:**

4: Strong: This paper deepens the understanding of some phenomenon or lowers the barriers to an existing research direction.

**Paper Topic And Main Contributions:**

This paper introduces a contrastive learning-based watermarking method for Bert-based Pre-trained Language Model (PLM) classification tasks. To enhance the robustness of this approach, a weighted perturbation strategy has been designed. The effectiveness of the proposed methods is demonstrated through the use of multiple baselines and evaluation metrics.

**Reasons To Accept:**

1. The motivation behind the research questions addressed in this paper is clearly and comprehensively explained.
2. The working principles of the proposed contrastive learning-based watermarking method are well-described. Particularly, Figure 1 provides informative and straightforward insights.
3. The paper not only explores Rare Words as triggers but also discusses and conducts experiments involving combinations of common words.
4. Both the accuracy and the success rate of ownership verification are thoroughly evaluated for both the baselines and the proposed methods.

**Reasons To Reject:**

1. Certain key points remain unclear, such as the lack of explanation for the two loss functions in Equation 3.
2. There is a mix of notation for \theta_0 and \theta in Equation (1), which requires clarification and consistency.

**Reproducibility:**

4: Could mostly reproduce the results, but there may be some variation because of sample variance or minor variations in their interpretation of the protocol or method.

**Reviewer Confidence:**

4: Quite sure. I tried to check the important points carefully. It's unlikely, though conceivable, that I missed something that should affect my ratings.

---

> ### Author Rebuttal · Authors · 2023-08-24
>
> Thank you for pointing out these issues，we will make our notations clear in the revision.
>
> In Equaition(3), the $L_{MLM}$ represents the Masked Language Modeling pre-training objective loss, and $L_{2}$ represents the L2 norm loss.
> In Equation (1)，we use $\theta_0$ for vanilla PLM and $\theta$ for trained parameters respectively.

---

### Official Review · Reviewer_NT7H · 2023-08-10

**Soundness:** 3

**Excitement:**

4: Strong: This paper deepens the understanding of some phenomenon or lowers the barriers to an existing research direction.

**Paper Topic And Main Contributions:**

This paper is about watermarking pre-trained language (PLMs) model, which is a sub-class of the model watermarking method. The unique attack of weight-changing due to the fine-tuning nature of PLMs poses a big challenge to watermark PLMs. To solve this, they propose a backdoor-based watermarking method powered by contrastive learning, which aims to accommodate the watermarked samples in an isolated area in the embedding space. The results show a high success rate of identifying watermarks in regular cases and robustness to cases when watermark removal methods are applied.

**Questions For The Authors:**

There can be another realistic setting (attack) where the PLMs are distilled from a watermarked PLM. The reviewer is curious about how the proposed method is robust to such attack.

**Reasons To Accept:**

1. The proposed method considers a realistic and non-trivial setting where the identification of PLM can be detected after weight changing (i.e., fine-tuning) and with limited access (i.e., only access to input and output) to the protected PLM. The experimental outcomes observed in text classification tasks not only exhibit commendable performance in verifying ownership but also demonstrate nearly no detrimental impact on downstream task performance.

2. The concept of backdoor-based watermarking and the manipulation of embedding space present a propitious avenue in the realm of watermarking PLMs where the watermarked models are susceptible to attacks involving alterations to their weights, often referred to as fine-tuning.

**Reasons To Reject:**

1. The experimented pre-trained models (PLM) mainly use the masked language model (MLM) pre-training objective, which may result in similar patterns in embedding spaces. It is interesting to see how the proposed method performs on more diverse BERT-like PLMs such as DeBERTa (decoding enhanced BERT) and ELECTRA (pretrain with an auxiliary discriminative model).

2. The method is only evaluated on text classification tasks.

**Reproducibility:**

4: Could mostly reproduce the results, but there may be some variation because of sample variance or minor variations in their interpretation of the protocol or method.

**Reviewer Confidence:**

4: Quite sure. I tried to check the important points carefully. It's unlikely, though conceivable, that I missed something that should affect my ratings.

---

> ### Author Rebuttal · Authors · 2023-08-25
>
> Thank you for your valuable comments.
>
> **Q1:The experimented pre-trained models (PLM) mainly use the masked language model (MLM) pre-training objective, which may result in similar patterns in embedding spaces. It is interesting to see how the proposed method performs on more diverse BERT-like PLMs such as DeBERTa (decoding enhanced BERT) and ELECTRA (pretrain with an auxiliary discriminative model).**
>
> R1:This is a very interesting question! To answer it, we conducted additional experiments on DeBERTa-base and ELECTRA-base using IMDB. The experimental results are provided below.
> | Model  | ACCU  | OVSR  |
> |---|---|---|
> | vanilla-DeBERTa     | 95.80.   |   0.00 |
> | watermarked-DeBERTa     | 95.56.  | 100.00 |
> |  vanilla-ELECTRA   |   95.63. |  0.00 |
> | watermarked-ELECTRA| 95.30| 100.00|
>
> Among them, "vanilla" represents the model that has not been injected with watermark, while "watermarked" represents the model that has been injected with watermark using our method. The experimental results indicate that even on more complex and diverse pre-trained language models  such as DeBERTa and ELECTRA, our method can still achieve excellent results. In fact, this is expected because our watermark injection method is based on backdoors, and the backdoor learning process is based on the over-parameterization of PLMs. That is, PLMs learn pre-training tasks themselves with redundant parameters (which is also the cornerstone of model pruning and model distillation) that can be used for backdoor learning. This means that the complexity of the pre-training task itself does not affect the implantation of backdoors. Therefore we expect that our method can successfully inject watermark even on models with more complex pre-training tasks and different embedding spatial patterns.
>
> **Q2:The method is only evaluated on text classification tasks.**
>
> R2:This is a good question. Watermarking on the generative model like GPT is an urgent need due to the bloom of LLMs in recent months. Unfortunately, it is a different setting due to the very different nature of the generation task and the classification task. In this study we mainly focus on the classification task and we plan to work on how to watermark a generative model in the future.
>
> **Q3:There can be another realistic setting (attack) where the PLMs are distilled from a watermarked PLM. The reviewer is curious about how the proposed method is robust to such attack.**
>
> R3:This study discusses the most common paradigm, which is the pre-training then finetuning paradigm. Since distillation generally affects performance compared to finetuning, this study mainly focuses on the most common paradigm. Whether watermarks can be maintained under distillation is a question worth exploring, but it is a new scenario and out of our scope. We hope to explore this question based on this study in the future.

---

### Official Review · Reviewer_j5wj · 2023-08-13

**Soundness:** 3

**Excitement:**

2: Mediocre: This paper makes marginal contributions (vs non-contemporaneous work), so I would rather not see it in the conference.

**Missing References:**

Yutong Wu, et al. "Watermarking Pre-trained Encoders in Contrastive Learning"
Tainxing Zhang, et al. "Awencoder: Adversarial watermarking pre-trained encoders in contrastive learning"
Xiaoyi Chen et al. "Apple of Sodom: Hidden Backdoors in Superior Sentence Embeddings via Contrastive Learning"

**Paper Topic And Main Contributions:**

In this paper, the author introduces a watermark framework tailored for pre-trained language models, with a particular emphasis on BERT-based models. The proposed framework adopts a backdoor-based watermarking technique, which embeds a secret function into the
 protected model. The model's owner can then leverage this secret function for verification purposes. At the core of this approach is a secret function wherein a specific trigger word forces all input samples to map to an identical embedding. Thus, samples injected with this trigger word are highly likely to be categorized into the same class. To achieve this, the author employs a method rooted in contrastive learning. To enhance the robustness of the watermark, the paper delves into a weight perturbation training strategy. The efficacy of the proposed framework is demonstrated through experiments on basic classification tasks, such as SST2 and IMDB.

**Questions For The Authors:**

Please refer to my Reasons To Reject

**Reasons To Accept:**

1. The structure of the paper is clear and logically organized, making it easy for readers to follow.
2. The author has assessed the model against multiple adaptive attacks and demonstrated commendable performance.


**Reasons To Reject:**

1. The concept of utilizing latent embeddings for watermarking is not a novel one. Numerous studies in both computer vision and NLP domains have broached this subject. For example, Yutong Wu et al.'s "Watermarking Pre-trained Encoders in Contrastive Learning" explored a very similar idea. Additionally, the method proposed in this paper does not seem to offer any insights uniquely tailored for PLMs. The proposed framework can also apply to computer vision encoders without significant modification. Additionally, the idea behind weighted perturbation seems to be borrowed from prior research. I find myself questioning the core contribution of this work. It would be beneficial if the author could delineate the distinct contributions of this paper more explicitly.
2. The evaluation of the proposed methods is limited to a few toy datasets. I wonder how the framework might fare on more extensive and complex datasets.

**Reproducibility:**

4: Could mostly reproduce the results, but there may be some variation because of sample variance or minor variations in their interpretation of the protocol or method.

**Reviewer Confidence:**

4: Quite sure. I tried to check the important points carefully. It's unlikely, though conceivable, that I missed something that should affect my ratings.

---

> ### Author Rebuttal · Authors · 2023-08-23
>
> Thank you for your valuable comments.
>
> **Q1:”The concept of utilizing latent embeddings for watermarking is not a novel one. Numerous studies in both computer vision and NLP domains have broached this subject”**
>
> R1:Although watermarking with latent embedding has been mentioned in previous research, how to inject the watermark into the PLM and make it robust to finetuning is still an unexplored problem, especially when the downstream task is unknown. In addition, we propose to use weight perturbation-based methods to improve the robustness of the watermark for possible removal techniques.
>
> **Q2:”Additionally, the method proposed in this paper does not seem to offer any insights uniquely tailored for PLMs. The proposed framework can also apply to computer vision encoders without significant modification”**
>
> R2: How to watermark a computer vision is out of our scope, we aim to protect the Intellectual Property of PLMs. Besides, there are differences between watermarking a computer vision encoder and a PLM. (e.g. the choose of trigger, in our study, we choose to use a rare word or a combination of common words as the trigger)
>
> **Q3:”Additionally, the idea behind weighted perturbation seems to be borrowed from prior research. I find myself questioning the core contribution of this work. It would be beneficial if the author could delineate the distinct contributions of this paper more explicitly.”**
>
> R3: Weight perturbation is a training trick which can be applied in multiple scenarios (e.g. improving the adversarial robustness or generality). We are the first study to apply weight perturbation to model watermarking, and our experiments demonstrate that doing so can improve the robustness of model watermarking against some watermark removal attacks such as model pruning.
> 	Generally, our main contributions are:
> 	1.We explore how to watermark PLMs without prior knowledge about downstream datasets, we propose a novel framework to do that, which includes how to inject the watermark and how to verify the existence the watermark.
> 	2.Based on contrastive learning and weight perturbation, we improve the robustness of the watermark against common watermark removal attacks.
>
> **Q4:The evaluation of the proposed methods is limited to a few toy datasets. I wonder how the framework might fare on more extensive and complex datasets.**
>
> R4: The main motivation of our method is that we can claim ownership of the PLM without any prior knowledge of the downstream dataset, so theoretically it can achieve satisfactory results on larger and more complex datasets. Related work also tends to use datasets that are also toy in size, as the downstream dataset size and complexity are not a concern for them (e.g. Yutong Wu, et al. "Watermarking Pre-trained Encoders in Contrastive Learning" uses STL10 and GTSRB as their downstream datasets, which has 113000 and 51800 images respectively).

---

### Meta-Review · Area_Chair_cYZv · 2023-09-14

**Recommendation:** 4

**Metareview:**

This works proposes a method to embed watermarks into neural networks in order to protect their intellectual property. Employing contrastive learning, the authors create backdoors in the form of rare or combined words that can trigger specific labels at inference. Crucially, the proposed approach resists model finetuning on unknown datasets, as well as some watermark removal attacks that the authors evaluated.

Reviewers have all acknowledge the technical quality, clarity and readability of the paper, while opinions on its novelty and significance vary. On the one hand, the proposed method addresses a clear, well-motivated problem and achieves good results. On the other hand, the approach is restricted to non-generative language models, limiting its reach; and though it resists some common attacks in those evaluation settings, whether it withstands more aggressive or adaptive evaluation or a wider set of tasks, models and datasets is not certain. Questions of the novelty of the approach have also been raised with respect to the method and modality: overall, the authors have convincingly argued that their use of contrastive learning differs from other works, and leverages text-specific elements in backdoors.

---

### Decision · Program_Chairs · 2023-10-07

**Decision:**

Accept-Findings

**Comment:**

This works proposes a method to embed watermarks into neural networks in order to protect their intellectual property. Employing contrastive learning, the authors create backdoors in the form of rare or combined words that can trigger specific labels at inference. Crucially, the proposed approach resists model finetuning on unknown datasets, as well as some watermark removal attacks that the authors evaluated.

Reviewers have all acknowledge the technical quality, clarity and readability of the paper, while opinions on its novelty and significance vary. On the one hand, the proposed method addresses a clear, well-motivated problem and achieves good results. On the other hand, the approach is restricted to non-generative language models, limiting its reach; and though it resists some common attacks in those evaluation settings, whether it withstands more aggressive or adaptive evaluation or a wider set of tasks, models and datasets is not certain. Questions of the novelty of the approach have also been raised with respect to the method and modality: overall, the authors have convincingly argued that their use of contrastive learning differs from other works, and leverages text-specific elements in backdoors.